# Carbon Emission Factor Multi-time Scale Prediction with Adaptive Graph Convolution Strategy

Songyan Wang[1], Xiongfeng Ye[2], Wei Wang[2], Xuejun Jiang[1,3]*, Qinmin Yang[1,3]

1. College of Control Science and Engineering, Zhejiang University, Hangzhou, China
2. Quzhou Power Supply Company, State Grid Zhejiang Electric Power Co., Ltd., Quzhou, China
3. Huzhou Institute of Zhejiang University, Huzhou, China
22260155@zju.edu.cn,734421480@qq.com,25100947@qq.com,jiangxuejun@zju.edu.cn,qmyang@zju.edu.cn

*Abstract*—The incremental attainment of the dual carbon objective has prompted a growing emphasis on precision carbon management as a significant area of concern. The carbon emission factor, serving as a bridge between electricity consumption and carbon emissions, aids grid operators in analysis and management. However, its complex characteristics make it challenging to predict, hindering grid managers in planning future electricity consumption. To address this issue, we introduce CEFPNet, a framework designed to obtain carbon emission factor prediction results by extracting correlations between variables and temporal characteristics at different time scales. In CEFPNet, the multivariate time series (MTS) input first exposes its features through the embedding layer, and then enters multiple G2CBlocks with residual connections to extract features and obtain prediction results. Within each G2CBlock, we apply the Fast Fourier Transform (FFT) to identify different time scales, segment the sequence into a three-dimensional tensor, and perform graph convolution to capture inter-variable relationships. Subsequently, multiple two-dimensional convolutions are applied to extract time series information, which is then reshaped back into a two-dimensional tensor to produce the final prediction. Results from three real datasets show that CEFPNet performs better than comparison methods in predicting the carbon emission factor. In addition, ablation experiments are delivered to prove the effectiveness of CEFPNet's substructures.

*Index Terms*—carbon emission factor, multi-time scale prediction, adaptive graph convolution, multivariate time series

## I. Introduction

In recent decades, global population and economic growth have led to a sharp increase in energy demand, drawing significant attention to the subsequent climate deterioration. As the world's second-largest economy and largest carbon emitter, China contributes around 27% of global carbon emissions, with over 85% stemming from energy consumption [1]. In response, China has implemented a dual-carbon strategy to address resource and environmental challenges. Presently, China's power system contributes over 40% of the nation's total carbon emissions, making it the largest carbon-emitting industry. Achieving carbon neutrality in the power sector is pivotal to the "3060" goal, with the sector bearing a substantial responsibility for carbon emission reduction.

*Corresponding author

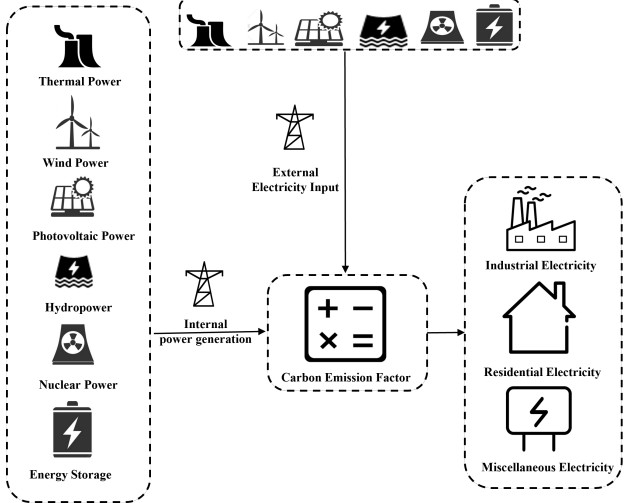

Fig. 1. Carbon emission factor calculation diagram

The carbon emission factor is a crucial link between electricity consumption and carbon emissions, effectively transferring carbon responsibility from the power generation side to the electricity consumption side. As per the IPCC's methodology for carbon measurement emission factors, the carbon emissions from electricity consumption can be calculated using the following formula:

$$CE = factor \times E \qquad (1)$$

where $CE$ is carbon emissions, $factor$ denotes the electricity consumption-side carbon emission factor, $E$ stands for electricity consumption.This article abbreviates the electricity consumption-side carbon emission factor as the electricity carbon emission factor. It signifies the amount of carbon emissions generated per kilowatt-hour of electricity consumed. The carbon emission factor, depending on the source of electricity, is calculated on the consumption side using the power flow tracing algorithm. The calculation framework is shown in the Fig. 1. Existing electricity carbon emission factors are categorized into national grid electricity carbon emission factors, regional electricity carbon emission factors,

and provincial electricity carbon emission factors based on measurement dimensions. However, these electricity carbon emission factors have coarse granularity, which limits their utility in power grid dispatching. Thus, predicting electricity carbon emission factors in smaller areas becomes imperative. This prediction not only aids power system planners in optimizing energy planning but also facilitates the consumption of new energy sources and enhances energy supply security.

Currently, carbon emission factors prediction encounters several challenges. Firstly, The carbon emission factor on the consumption side is determined by the 220kV and 110kV power grids' carbon emission factor, while on the generation side, it correlates with various energy sources and user electricity consumption. Hence, it is imperative to analyze the intricate interactions among multiple variables. Secondly, The interactions among variables exhibit variations across different time scales. On shorter time scales, higher electricity consumption will lead to insufficient new energy consumption, causing a rise in the electricity carbon emission factor. Conversely, on longer time scales, sufficient new energy consumption mitigates this effect, resulting in a more stable electricity carbon emission factor. Finally, Since the electricity carbon emission factor essentially reflects the carbon content in electricity, it is imperative to consider new energy power generation. Nevertheless, the significant fluctuations in the output of new energy power generation pose challenges to accurately predicting its quantity, thereby complicating the prediction of the electricity carbon emission factor.

Due to the above challenges, it is difficult to accurately predict the electricity carbon emission factor using existing models. In recent years, many prediction models have been proposed and used to solve practical problems, such as energy [2], finance [3], etc. With the development of deep learning, many deep learning models model the temporal information of series to complete predictions, such as [4]–[7]. Transformer-based models are extensively employed owing to their robust modeling capabilities and generalization capabilities [8]–[10].

In addition to temporal characteristics, comprehending the intricate relationships among multivariate variables is crucial for multivariate time series (MTS) prediction. In particular, dimensional correlations across various time scales offer valuable insights for enhancing predictions. Reference [11] integrates a graph learning module to autonomously extract directional relationships among variables. Reference [12] models the dependencies between different variables, completes the extraction of dimensional information and temporal information, and realizes the prediction of MTS. Many scholars have also studied the issue of uncertainty in the output of new energy power generation. Reference [10] believes that real-world data are non-stationary, and this non-stationary will affect the learning ability of the model. To address this, it preprocesses the input sequence and improves the attention mechanism, resulting in a significant enhancement of the model's prediction performance.

Accurate forecasts are critical to grid operations, with timescales ranging from hours to days. Grid operators utilize the electricity carbon emission factor to steer the electricity consumption behavior of consumers, thereby reducing carbon emissions and mitigating the challenges associated with new energy power consumption.

Based on the above literature, this paper proposes a novel network structure. This model considers the non-stationarity of the series, not only extracts the temporal information of the series but also extracts intricate relationship information between variables at different time scales. The main contributions of this paper are as follows:

1 We conducted comprehensive research on existing MTS prediction models and found that they inadequately address the challenges associated with predicting electricity carbon emission factors, such as non-stationary data and complex relationships between variables at different time scales.

2 In this paper, we present xxx, a model that addresses both the non-stationarity of series and effectively integrates temporal and dimensional characteristics for MTS prediction. Our model can capture variable correlations at different time scales and effectively identify the periodicity and trends within the series.

3 We conduct experiments on three datasets, including the electricity carbon emission factor, to demonstrate the superiority of the proposed module and to validate the model's generalization performance.

## II. RELATED WORKS

Currently, there are few studies on the prediction of carbon emission factors. Most research efforts have concentrated on accurately and reasonably calculating carbon emission factors, employing various methods to derive it from different perspectives. Although one application of carbon emission factors is to precisely measure carbon emissions, there is limited research on using it for carbon emissions prediction. Typically, carbon emissions are predicted directly without considering the carbon emission factors. Additionally, existing research on carbon emission prediction often employs outdated methods that fail to account for a range of internal factors and mechanisms influencing carbon emissions, leading to inaccurate predictions. In recent years, however, many machine learning models have been successfully applied to multivariate time series prediction tasks, achieving promising results.

**Multivariate Time Series Forecasting**. Time series forecasting models are mainly divided into statistical models and machine learning models. Statistical models rely on rigorous mathematical derivations, presenting relationships within time series data in a parameterized form. This allows for a deep understanding of the data generation process, leading to effective predictions. Examples include autoregressive models, and autoregressive moving average models [13]. Machine learning models, particularly deep learning models, excel in complex time series predictions due to their ability to capture and model intricate nonlinear relationships. Long Short-Term Memory(LSTM) addresses the gradient vanishing and exploding problems faced by Recurrent Neural Network(RNN)

when processing long sequences by introducing a gating mechanism, thus performing well with long-time series data. However, LSTM has high computational complexity and long training time. LSTNet combines Convolutional Neural Network(CNN) and LSTM structures; CNN captures local patterns and seasonal characteristics, while LSTM captures long-term dependencies, enabling the model to process information on different time scales and extract data features more effectively. Temporal Convolutional Networks(TCN) employ a series of one-dimensional convolutional layers to capture local patterns and long-term dependencies, offering higher parallelism, computational efficiency, and strong generalization capabilities. Although TCN uses dilated convolution to increase the receptive field of the convolutional layer, it may still struggle with time series that have multi-scale features.

**Transformers for MTS Forecasting**. In 2017, the Transformer [4], proposed by the Google Machine Translation team, made a significant breakthrough in natural language processing by leveraging the self-attention mechanism and encoder-decoder structure. The self-attention mechanism of the Transformer enables it to capture information of any length and perform global modeling, making it applicable in the field of time series prediction. However, the Transformer's space complexity increases quadratically with the sequence length, posing challenges for practical applications.To address these shortcomings, Zhou introduces Informer [9], which utilizes a probabilistic sparse attention mechanism. The attention distillation mechanism and the design of the generative decoder enable the model to reduce memory usage during training and mitigate cumulative error. In 2022, the Alibaba Dharma Institute invented FEDformer [14], a frequency-enhanced Transformer, leveraging the fact that time series often have sparse representations in the frequency domain.

Transformer family models utilize the self-attention mechanism to extract temporal information. In 2022, a scholar [10] argued that time series data lack point-to-point semantic correlation, having only temporal relationships between consecutive points, and thus the self-attention mechanism might result in the loss of temporal information. Moreover, simple linear models have outperformed Transformer-based models in time series prediction, prompting reconsideration of the suitability of the attention mechanism for these tasks. In 2023, TimesNet [6] transformed one-dimensional time series data into a two-dimensional tensor based on multiple cycles, thereby expanding temporal variations into a two-dimensional space. This approach enabled the extraction of intra- and inter-cycle information, overcoming the representational limitations of one-dimensional time series.

The stationarity of data is a crucial factor when predicting time series. A stationary time series has a stable data distribution, making it easier for models to capture inherent patterns and trends, thus producing more accurate predictions. However, most real-world data are non-stationary. To address this issue, autoregressive integral moving average models(ARIMA) [15] transform non-stationary time series into stationary ones through differencing, eliminating trends and seasonal factors.

Non-stationary transformer [10] converts the input sequence into stationary data through the series stationarization module, but this will lead to the loss of feature information, so De-stationary Attention is used to extract features from non-stationary data, and finally restore the non-stationarity of the sequence to obtain the prediction results.

## III. METHODOLOGY

### A. Problem Formulation

This paper focuses on the multivariate time series forecasting task, where the objective is to predict future values of a time series $\hat{X}_{t:t+T} \in \mathbb{R}^{T \times D}$ given an input sequence $X_{t-L:t} \in \mathbb{R}^{L \times D}$. Here, $D$ represents the number of variables, $X_\tau^i$ represents a retrospective observation window containing the values of variable $i$ at the $\tau$ time point, ranging from $t-L$ to $t-1$, $L$ represents the size of the retrospective window and $T$ indicates the forecast time step.

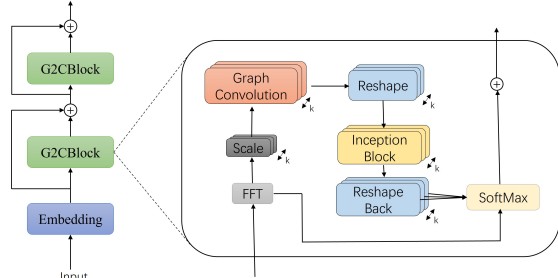

Fig. 2. overall architecture.

### B. Model Architecture

This paper designs an MTS prediction model based on the characteristics of the electric carbon factor to complete the prediction task. CEFPNet is a novel framework that aims to capture the time series characteristics and the correlation between series at different time scales. The overall model architecture is shown in the figure. CEFPNet first processes the input through an embedding layer and then connects multiple G2CBlocks with residuals. G2CBlock is the feature extraction module we proposed, designed to seamlessly integrate various components. Each G2CBlock consists of four steps: 1) Identify the time scale of the multivariate series; 2) Use adaptive graph convolution blocks to extract scale-related inter-sequence correlations; 3) Use two-dimensional convolution to extract time series characteristics, including periodicity and trend; 4) Use the Softmax function to adaptively aggregate information at different time scales. The entire process can be expressed by the following formula:

$$X^l = G2CBlock(X^{l-1}) + X^{l-1} \qquad (2)$$

where $X^l \in \mathbb{R}^{d_{model} \times L}$ represents the output of the $l$-th layer. $G2CBlock$ represents the operations and computations that constitute the core functionality of the CEFPNet layer. Initially, we set $X^0 = \mathbf{X}_{emb}$ represents the original input projected by the embedding layer to the deep features.

## C. Input Embedding

Given the uncertainty of renewable energy output, the data exhibits strong non-stationarity, making it difficult for the model to learn. Therefore, we first normalize the input series. The process can be expressed as follows:

$$\mu_{\mathbf{x}} = \frac{1}{L} \sum_{i=1}^{L} x_i, \sigma_{\mathbf{x}}^2 = \frac{1}{L} \sum_{i=1}^{L} (x_i - \mu_{\mathbf{x}})^2, \overline{X} = \frac{1}{\sigma_{\mathbf{x}}} \odot (x_i - \mu_{\mathbf{x}})$$
(3)

Where $x_i$ represents the input sequence, $L$ denotes the length of the input series, $\mu_{\mathbf{x}}$ represents the mean of the input sequence, $\sigma_{\mathbf{x}}^2$ represents the variance of the input sequence, and $\odot$ represents element-wise multiplication to obtain the normalized output $\overline{X}$.

Inspired by Informer [9], we embed $N$ variables into a vector of size $d_{model}$, where $X_{emb} \in \mathbb{R}^{d_{model} \times L}$. The embedding process is described as follows:

$$\mathbf{X}_{\text{emb}} = \alpha \operatorname{Conv} 1D \left( \overline{X} \right) + \mathbf{PE} + \sum_{p=1}^{P} \mathbf{SE}_p$$
(4)

We project $\overline{X}$ into a $d_{model}$-dimensional matrix using 1-D convolutional filters. The parameter $\alpha$ acts as a balancing factor, adjusting the size between the scalar projection and the local/global embeddings. $\mathbf{PE}$ represents the positional embedding of the input $\overline{X}$, $\mathbf{SE}_p$ is the learnable global timestamp embedding, and $\mathbf{X}_{\text{emb}}$ represents the output of the embedding layer.

## D. G2CBlock

Our objective is to utilize the G2CBlock to extract correlations and temporal characteristics of sequences across various time scales, thereby enhancing prediction accuracy. This is crucial as interactions between sequences can vary significantly with different time scales. For instance, at a smaller time scale, the carbon emission factor may rapidly decrease with an increase in renewable energy generation, while at a larger time scale, these variables exhibit different correlation patterns. Initially, we analyze the time scales. Inspired by TimesNet [6], we employ a Fast Fourier Transform to derive the different time scales of the multivariate time series. The process can be expressed as follows:

$$\mathbf{A} = \operatorname{Avg} \left( \operatorname{Amp} \left( \mathbf{FFT} \left( \mathbf{X}_{\text{emb}} \right) \right) \right),$$
$$f_1, \cdots, f_k = \operatorname{argTopk}(\mathbf{A}), p_i = \frac{L}{f_i}.$$
(5)

In this context, $\mathbf{FFT}()$ and $\operatorname{Amp}()$ denote the calculations of the Fast Fourier Transform and amplitude values, respectively. $\mathbf{A}$ represents the amplitude of each frequency, calculated as the mean across $d_{model}$ dimensions. To account for the sparsity of the frequency domain and to avoid high-frequency noise, we select only the top $K$ most significant frequencies as our time scales. Consequently, $k$ frequencies and their corresponding periods are obtained. Using these periods, we reshape the input

into a 3D tensor. This process is expressed by the following equation:

$$\mathcal{X}^i = \operatorname{Reshape}_{p_i, f_i} \left( \operatorname{Padding} \left( \mathbf{X}_{\text{in}} \right) \right), \quad i \in \{1, \ldots, k\},$$
(6)

Where $\mathbf{X}_{\text{in}}$ represents the input of the G2CBlock, and $\mathcal{X}^i$ represents the series reshaped based on time scale $p_i$.

To extract correlations between series at different time scales, inspired by MSGNet [16], we use graph convolution to identify dependencies. First, we reproject the tensor corresponding to time scale $p_i$ back to a tensor of $N$ variables, where $N$ represents the number of variables. Then, we employ two learnable matrices to derive an adaptive adjacency matrix, followed by the Mixhop graph convolution method [17] to capture the correlations between sequences. The entire process is expressed as follows:

$$\mathcal{H}^i = \mathbf{W}^i \mathcal{X}^i$$
$$\mathbf{M}^i = \operatorname{SoftMax} \left( \operatorname{ReLu} \left( \mathbf{E}_1^i \left( \mathbf{E}_2^i \right)^T \right) \right).$$
$$\mathcal{H}_{\text{out}}^i = \sigma \left( \|_{j \in \mathcal{P}} \left( \mathbf{M}^i \right)^j \mathcal{H}^i \right)$$
(7)

Where $\mathbf{W}^i$ represents a learnable weight matrix customized for the time scale $p_i$, $\mathbf{E}_1^i$ and $\mathbf{E}_2^i$ represent two learnable matrices, and $\mathbf{M}^i$ represents an adaptive adjacency matrix. $\sigma()$ represents an activation function, $\mathcal{P}$ represents a set of integer adjacency powers, $\|$ represents a column-level connection, connecting the intermediate variables generated during each iteration, and $\mathcal{H}_{\text{out}}^i$ represents the learned sequence relationship at the time scale $p_i$. Then, we use a multi-layer perceptron to project $\mathcal{H}_{\text{out}}^i$ back into a 3D tensor $\overline{\mathcal{X}}^i$.

To extract temporal information, we employ an Inception Block composed of multiple two-dimensional convolutions to capture periodicity and trends. Each column of $\overline{\mathcal{X}}^i$ represents data points at different times within a cycle under the time scale $p_i$, while each row represents data points at the same time across cycles under the time scale $p_i$. Thus, we can effectively extract temporal information using the Inception Block. The entire process can be expressed as follows:

$$\overline{\mathcal{X}}_{out}^i = Inception(\overline{\mathcal{X}}^i)$$
(8)

Where $\overline{\mathcal{X}}_{out}^i$ represents the dimensional information and temporal information at different time scales. Subsequently, we reshape the information from $K$ different time scales back into a two-dimensional matrix and then aggregate the information according to its amplitude values at different frequencies. The entire process is as follows:

$$\overline{\mathbf{X}}_{\text{out}}^i = \operatorname{Trunc} \left( \operatorname{Reshape}_{d_{model}, (p_i \times f_i)} \left( \overline{\mathcal{X}}_{out}^i \right) \right),$$
$$a_1, \cdots, a_k = \operatorname{SoftMax} \left( \mathbf{A}_{f_1}, \cdots, \mathbf{A}_{f_k} \right),$$
$$\overline{\mathbf{X}}_{\text{out}} = \sum_{i=1}^{k} a_i \overline{\mathbf{X}}_{\text{out}}^i.$$
(9)

We use linear projection to make predictions, followed by an inverse normalization module to restore the data's original non-stationarity. The process is illustrated in the following formula:

$$\overline{\mathbf{X}}_{t:t+T} = \mathbf{W_n}\overline{\mathbf{X}}_{\text{out}}\,\mathbf{W_t} + \mathbf{b},$$
$$\hat{\mathbf{X}}_{t:t+T} = \sigma_{\mathbf{x}} \odot \left(\overline{\mathbf{X}}_{t:t+T} + \mu_{\mathbf{x}}\right) \tag{10}$$

Where $\mathbf{W_t}$ , $\mathbf{W_n}$ , and $b$ are learnable parameters. The $\mathbf{W_t}$ matrix executes linear projection along the variable dimension, and $\mathbf{W_n}$ matrix executes linear projection along the time dimension. The resulting $\hat{\mathbf{X}}_{t:t+T}$ is the forecasted data.

## IV. EXPERIMENTS

**Datasets** To demonstrate the predictive capability and generalization ability of the proposed model for the carbon emission factor, we conducted experiments on three datasets: the Electric Carbon Factor (ECF) dataset, the Wind Power Generation (WPG) dataset, and Appliances Energy Prediction (AEP) dataset. The electric carbon factor dataset includes data on the electric carbon factors of the 220KV and 110KV sides, as well as regional electricity consumption, photovoltaic power generation, and waste power generation. The wind power generation dataset includes information on wind speed, wind direction, and generator speed. Appliances Energy Prediction Data is a public dataset that contains data on room temperature, humidity, and electricity consumption. Following standard protocol [9], we divided all datasets into training, validation, and test sets in a 7:1:2 ratio.

Due to dimensionality issues, we initially standardized the carbon factor emission dataset before conducting the analysis. As depicted in Figure 3(a)'s correlation analysis, there exists a notably high correlation among variables within the dataset. Particularly significant correlations are observed between the carbon emission factors of 220 KV and 110 KV, and the power consumption side, as well as between photovoltaic power generation and the electric carbon factor concerning power consumption. Thus, it is imperative to account for these inter-variable correlations. Additionally, Figure 3(b) illustrates that at a larger time scale, power consumption and user-side carbon emission factor exhibit a positive correlation. However, at a smaller time scale, owing to decreased new energy power generation, power consumption and user-side electric carbon factor display a negative correlation. Thus, studying variable correlation across different time scales is essential. Moreover, Figure 3(c) indicates evident periodicity among variables, with power consumption showing a pronounced trend. Consequently, it is crucial to extract the temporal characteristics of the variables. Therefore, we utilize the proposed G2CBlock to extract the aforementioned information.

**Experimental Setups** The experiments were conducted using an NVIDIA GeForce RTX 3090 24GB GPU, with Mean Squared Error (MSE) as the training loss function. The review window size for all models was set to $L = 96$ for a fair comparison, and the prediction lengths were $T = \{24, 48, 96, 288, 672\}$. These settings were applied uniformly across all models. The initial learning rate was set to $LR = 0.0001$, with a batch size of $Batch = 32$ and the number of epochs was $Epochs = 10$. Early stopping was utilized where applicable. The data was split into training, validation, and test sets in the ratios 0.7, 0.1, and 0.2, respectively.

**Baselines** we use the following popular MTS prediction models as baselines: 1) DLinear [10] , 2) Crossformer [12] , 3) Autoformer [8] , 4) Transformer [4] , 5) Informer [9].

**Metrics** The evaluation indicators used for the prediction model are Mean Squared Error (MSE) and Mean Absolute Error (MAE) [18]. The formula is as follows:

$$MSE = \frac{1}{N}\sum_{i=1}^{N}\left(\chi_i - \hat{\chi}_i\right)^2$$
$$MAE = \frac{1}{N}\sum_{i=1}^{N}|\chi_i - \hat{\chi}_i| \tag{11}$$

### A. Performance Comparison

For multivariate long-term sequence forecasting, CEFPNet achieves state-of-the-art performance across all benchmarks and all prediction length settings (Table I). Compared with the previous best model, NTDformer achieves a 10.2% average MSE reduction. Especially, under the input-96predict-672 setting, the MSE of CEFPNet is reduced by 18% in the electricity carbon factor dataset. Under the input-96predict-288 setting, the MSE of CEFPNet is reduced by 38% in the wind power dataset, reducing the MSE by 24.9% on average. Under the input-96predict-24 setting, the MSE of CEFPNet is reduced by 26% in the Appliances Energy Prediction Data, reducing the MSE by 16% on average. To sum up, NTDformer can not only handle the prediction task of carbon emission factor better but also has good results in the prediction of other data.

### B. Ablation Studies

Our approach focuses on extracting relationships between variables and complex time series information at different time scales. To verify the effectiveness of our proposed CEFPNet for MTS forecasting, we considered two ablation methods and verified them on three datasets. The variants of its implementation are explained below:

1 **w/o-AdapG:** We removed the adaptive graph convolution from the model and used only the Inception Block, composed of multiple two-dimensional convolutions, to extract time series information at different scales for prediction.

2 **w/o-MG:** We removed the multi-scale graph convolution from the model and used only the multi-head self-attention mechanism to account for the relationships between variables.

3 **w/o-IB:** We removed the Inception Block, which consists of multiple two-dimensional convolutions, from the model and used only multi-scale graph convolution to extract correlations between multiple variables at different time scales for generating prediction results.

Table II presents the results of the ablation experiment. Specifically, when the multi-scale graph convolution was removed, the model performance dropped significantly, indicating that extracting the relationships between multiple variables is crucial for predicting multivariate time series. Similarly, replacing the multi-scale graph convolution with a multi-head self-attention mechanism also led to a significant performance drop, highlighting the importance of capturing interactions between variables at different time scales. Lastly,

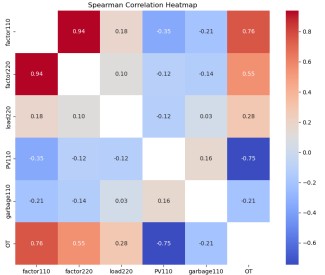

(a) Spearman correlation coefficient of carbon emission factor

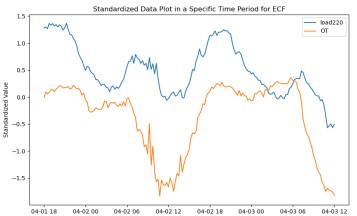

(b) Time Series Plot of the Data PV110

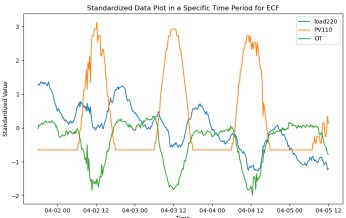

(c) Time Series Plot of the Data load

Fig. 3. the characteristics of carbon emission factor dataset

TABLE I
RESULTS WITH DIFFERENT PREDICTION LENGTHS.

| model | | Our model | | DLinear | | Crossformer | | Autoformer | | Transformer | | Informer | |
|---|---|---|---|---|---|---|---|---|---|---|---|---|---|
| Metric | | MSE | MAE | MSE | MAE | MSE | MAE | MSE | MAE | MSE | MAE | MSE | MAE |
| ECF | 24 | **0.349** | **0.350** | 0.373 | 0.382 | 0.358 | 0.357 | 0.784 | 0.586 | 0.751 | 0.621 | 0.800 | 0.629 |
| | 48 | **0.388** | **0.370** | 0.423 | 0.415 | 0.416 | 0.387 | 0.734 | 0.595 | 0.812 | 0.606 | 0.843 | 0.589 |
| | 96 | **0.411** | **0.385** | 0.439 | 0.428 | 0.473 | 0.411 | 0.739 | 0.588 | 0.928 | 0.686 | 0.833 | 0.619 |
| | 288 | **0.448** | **0.420** | 0.507 | 0.482 | 0.563 | 0.501 | 0.848 | 0.596 | 1.099 | 0.694 | 1.060 | 0.777 |
| | 672 | **0.504** | **0.466** | 0.616 | 0.556 | 0.679 | 0.583 | 0.990 | 0.667 | 1.162 | 0.775 | 1.294 | 0.839 |
| WPG | 24 | **0.014** | **0.055** | 0.017 | 0.086 | 0.024 | 0.095 | 0.064 | 0.189 | 0.033 | 0.107 | 0.037 | 0.106 |
| | 48 | **0.021** | **0.070** | 0.025 | 0.108 | 0.047 | 0.123 | 0.077 | 0.199 | 0.063 | 0.155 | 0.054 | 0.145 |
| | 96 | **0.034** | **0.091** | 0.048 | 0.158 | 0.085 | 0.178 | 0.139 | 0.233 | 0.139 | 0.233 | 0.124 | 0.249 |
| | 288 | **0.087** | **0.144** | 0.141 | 0.282 | 0.203 | 0.297 | 0.172 | 0.268 | 0.159 | 0.274 | 0.406 | 0.429 |
| | 672 | **0.218** | **0.216** | 0.287 | 0.392 | 0.364 | 0.388 | 0.408 | 0.383 | 0.397 | 0.413 | 0.637 | 0.514 |
| AEP | 24 | **0.353** | **0.355** | 0.478 | 0.453 | 0.512 | 0.455 | 0.618 | 0.554 | 0.815 | 0.606 | 0.836 | 0.642 |
| | 48 | **0.508** | **0.471** | 0.624 | 0.542 | 0.824 | 0.625 | 0.960 | 0.720 | 0.974 | 0.699 | 1.228 | 0.819 |
| | 96 | **0.672** | **0.540** | 0.825 | 0.653 | 1.062 | 0.737 | 1.000 | 0.725 | 1.437 | 0.844 | 1.761 | 1.011 |
| | 288 | **1.027** | **0.703** | 1.305 | 0.826 | 1.688 | 0.980 | 1.365 | 0.845 | 1.932 | 1.013 | 2.214 | 1.150 |
| | 672 | **1.718** | **0.932** | 1.863 | 1.037 | 2.306 | 1.167 | 1.879 | 0.987 | 2.436 | 1.177 | 2.512 | 1.203 |

TABLE II
ABLATION EXPERIMENT RESULTS ON ECF

| Models | CEFPNet | | w/o-AdapG | | w/o-MG | | w/o-IB | |
|---|---|---|---|---|---|---|---|---|
| Metric | MSE | MAE | MSE | MAE | MSE | MAE | MSE | MAE |
| 24 | 0.349 | **0.350** | 0.414 | 0.395 | **0.347** | 0.362 | 0.371 | 0.368 |
| 48 | **0.388** | **0.370** | 0.448 | 0.417 | 0.403 | 0.393 | 0.413 | 0.386 |
| 96 | **0.411** | **0.385** | 0.479 | 0.441 | 0.419 | 0.402 | 0.440 | 0.404 |
| 288 | **0.448** | **0.420** | 0.539 | 0.471 | 0.482 | 0.453 | 0.462 | 0.421 |
| 672 | **0.504** | **0.466** | 0.578 | 0.501 | 0.590 | 0.546 | 0.552 | 0.486 |

the removal of the Inception Block resulted in a notable decline in model performance, demonstrating the critical role of temporal information in multivariate time series prediction.

## V. CONCLUSIONS

In this paper, we introduce CEFPNet, a novel framework designed to address the limitations of existing models in predicting carbon emission factors. Our method employs periodicity as the time scale, utilizes graph convolution to capture correlations between variables, and applies multiple two-dimensional convolutions to capture temporal information, including periodicity and trends. Experiments on three real-world datasets demonstrate that CEFPNet outperforms existing models in electric carbon factor prediction accuracy and exhibits strong generalization performance. Furthermore,

the effectiveness of our proposed G2CBlock in capturing both the correlations and temporal nature of variables is validated.

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
