# OpenReview forum: "Carbon Emission Factor Multi-time Scale Prediction with Adaptive Graph Convolution Strategy"
_IEEE.org/ICIST/2024/Conference — IEEE ICIST 2024 Conference Submission_

### Official Review · Reviewer_kTHb · 2024-08-21
**accept**

**Rating:** 7
**Confidence:** 3

**Review:**

This paper proposes a novel network structure with carbon emission factor multi-time scale prediction , which demonstrated excellent performance. The theory is correct and can be accepted after responding the following comments.
(1)There are many typos and grammar errors. The authors should have a native English speaker or software packages to perform the editing check.
(2)Please consider selecting recent papers from journals or conferences in exchange for some older papers.
(3) The conclusion of the article suggests using the present perfect tense for description.

---

### Official Review · Reviewer_kC7z · 2024-08-22
**This paper introduced the CEFPNet framework, which aims to extract the correlation between variables and time features at different time scales to obtain carbon emission factor prediction results. The topic of this paper is interesting.**

**Rating:** 8
**Confidence:** 4

**Review:**

This paper introduced the CEFPNet framework, which aims to extract the correlation between variables and time features at different time scales to obtain carbon emission factor prediction results. The topic of this paper is interesting. Below is a list of comments that should be taken into account further when revising the paper.
1.	The contribution of this article should be compared with previous literature, and the basic technical difficulties of this article should be listed? And what methods should be used to solve this problem, emphasizing novelty and technological contribution.
2.	In the ablation studies section, in order to verify the effectiveness of the proposed CEFPNet for MTS prediction, two ablation methods were considered and should be elaborated on.
3.	The paper should provide a detailed description of the innovative points to enable readers to quickly understand the article. Meanwhile, please elaborate on the future plans.

---

### Official Review · Reviewer_EFz1 · 2024-08-25
**Accept**

**Rating:** 7
**Confidence:** 3

**Review:**

Comment: This paper introduces the CEFPNet framework, which aims to obtain carbon emission factor prediction results by extracting the correlation between variables and time features at different time scales and ablation experiments are delivered to prove the effectiveness of CEFPNet’s substructures. The theory is correct and can be accepted after responding the following comments.
(1)	What is the contribution of the paper? It should be highlighted both in the introduction and in the content.
(2) There are obvious grammar errors in the article. The authors should have a native English speaker or software packages to perform the editing check.
(3) The textual description in the experiment is too cumbersome.

---

### Decision · Program_Chairs · 2024-09-06

Accept (Oral)